# Myo-Inositol Moderates Glucose-Induced Effects on Human Placental ^13^C-Arachidonic Acid Metabolism

**DOI:** 10.3390/nu14193988

**Published:** 2022-09-26

**Authors:** Oliver C. Watkins, Victoria K. B. Cracknell-Hazra, Reshma Appukuttan Pillai, Preben Selvam, Hannah E. J. Yong, Neha Sharma, Sathya Narayanan Patmanathan, Amaury Cazenave-Gassiot, Anne K. Bendt, Keith M. Godfrey, Rohan M. Lewis, Markus R. Wenk, Shiao-Yng Chan

**Affiliations:** 1Department of Obstetrics and Gynaecology, Yong Loo Lin School of Medicine, National University of Singapore, Singapore 119228, Singapore; 2Singapore Institute for Clinical Sciences, Agency for Science, Technology and Research, Singapore 117609, Singapore; 3NIHR Southampton Biomedical Research Centre, University of Southampton and University Hospital Southampton NHS Foundation Trust, Southampton SO17 1BJ, UK; 4Department of Biochemistry and Precision Medicine TRP, Yong Loo Lin School of Medicine, National University of Singapore, Singapore 117596, Singapore; 5Singapore Lipidomics Incubator, Life Sciences Institute, National University of Singapore, Singapore 119077, Singapore; 6MRC Lifecourse Epidemiology Centre, University of Southampton, Southampton SO17 1BJ, UK; 7Institute of Developmental Sciences, Faculty of Medicine, University of Southampton, Southampton SO17 1BJ, UK

**Keywords:** lipid, pregnancy, LCMS, metabolism, diabetes

## Abstract

Maternal hyperglycemia is associated with disrupted transplacental arachidonic acid (AA) supply and eicosanoid synthesis, which contribute to adverse pregnancy outcomes. Since placental inositol is lowered with increasing glycemia, and since myo-inositol appears a promising intervention for gestational diabetes, we hypothesized that myo-inositol might rectify glucose-induced perturbations in placental AA metabolism. Term placental explants (*n* = 19) from women who underwent a mid-gestation oral glucose-tolerance-test were cultured with ^13^C-AA for 48 h in media containing glucose (5, 10 or 17 mM) and myo-inositol (0.3 or 60 µM). Newly synthesized ^13^C-AA-lipids were quantified by liquid-chromatography-mass-spectrometry. Increasing maternal fasting glycemia was associated with decreased proportions of ^13^C-AA-phosphatidyl-ethanolamines (PE, PE-P), but increased proportions of ^13^C-AA-triacylglycerides (TGs) relative to total placental ^13^C-AA lipids. This suggests altered placental AA compartmentalization towards storage and away from pools utilized for eicosanoid production and fetal AA supply. Compared to controls (5 mM glucose), 10 mM glucose treatment decreased the amount of four ^13^C-AA-phospholipids and eleven ^13^C-AA-TGs, whilst 17 mM glucose increased ^13^C-AA-PC-40:8 and ^13^C-AA-LPC. Glucose-induced alterations in all ^13^C-AA lipids (except PE-P-38:4) were attenuated by concurrent 60 µM myo-inositol treatment. Myo-inositol therefore rectifies some glucose-induced effects, but further studies are required to determine if maternal myo-inositol supplementation could reduce AA-associated pregnancy complications.

## 1. Introduction

### 1.1. Arachidonic Acid in Pregnancy

Arachidonic acid (AA), an omega-6 long-chained polyunsaturated fatty acid, is vital for fetal-placental development and long-term offspring health [1]. AA and its derivatives are critical regulators of placental function and parturition, but both placenta and fetus have limited capacity to synthesize AA and rely on maternal supply [2,3]. Since AA is lowly abundant in the maternal circulation, well-regulated and preferential placental AA uptake, metabolism and fetal transfer is essential to meet fetal-placental demand [1,4]. However, transplacental AA transfer appears dysregulated in women with type 1, type 2 and gestational diabetes, resulting in reduced fetal-cord circulating AA lipids [5,6,7].

Diabetes and in vitro glucose treatment also disrupt the placental synthesis of AA-derived signalling molecules, increasing the synthesis of eicosanoids like thromboxane B2, HETES and LTB4, but decreasing the synthesis of 6-keto prostaglandin F1a [8,9,10]. Many of these molecules act as intracellular, paracrine and endocrine signals to regulate uteroplacental processes, inflammation and vascular function, so alterations impact fetal nutrient supply [11]. Indeed, placental and cord AA-phospholipids; a major source of AA for the synthesis of such molecules, are positively associated with birthweight [12,13]. Dysregulation of AA and eicosanoid metabolism in utero may also increase the risk of preterm birth, since the labor process is also driven by many AA-derived signaling molecules [14].

### 1.2. Myo-Inositol as a Potential Treatment for Disordered Placental Lipid Metabolism

Myo-inositol, the predominant inositol isomer in mammals, is an important regulatory polyol, synthesized endogenously in humans and abundant in dietary grains and fruits [15,16]. In pregnancy, inositols and inositol-containing derivatives play critical roles in regulating endocrine and paracrine signaling (including insulin signaling), placental physiology, nutrient transport, and lipid metabolism, which impact fetal growth and development [17]. However, inositol processing and action are dysregulated in disorders involving insulin resistance, including gestational diabetes (GDM), where placental inositol content is reduced [17]. Myo-inositol supplementation has shown inconsistency in the prevention of GDM and fetal macrosomia [18,19], but shows promise in reducing pre-term births [19,20], where dysregulated in utero AA metabolism may be implicated. We have also previously found that endogenous myo-inositol may specifically suppress glycemia-induced promotion of birthweight [21]. Myo-inositol also appears important for neural development. In humans low maternal circulatory myo-inositol is associated with spina bifida [22], whilst in mouse dams myo-inositol supplementation reduced the frequency of mouse neural tube defects [23]. Glucose-induced neural tube defects in rodent embryos were also prevented by myo-inositol through the rectification of AA-derived eicosanoid synthesis [24,25].

### 1.3. Placental AA Metabolism

Placental AA-metabolism regulates the availability of AA for use by the placenta, fetus and chorio-amniotic membranes, and is critical for healthy fetal development and parturition. Hence, glucose-induced alterations in placental AA-metabolism may play a role in the pathophysiology of hyperglycemia-related complications of pregnancy, including fetal macrosomia and preterm birth [14,26]. Thus, if inositol can attenuate the effects of glycemia on placental AA metabolism, adequate placental inositol may be protective against hyperglycemia-associated pregnancy complications.

Our study is designed to measure the production of ^13^C-AA-labeled lipids by placental explants from exogenous ^13^C-labeled-AA. We hypothesized that glucose induces perturbations in placental AA lipid metabolism that can be rectified by myo-inositol treatment. We thus aimed to characterize variations in in vitro placental AA-lipid processing capacity in relation to gestational glycemia and BMI, to describe changes induced directly by glucose in vitro and assess the effects of concurrent myo-inositol treatment, as well as seek supportive evidence in a separate cohort of snap-frozen placenta.

## 2. Methods and Materials

Placentas from nineteen pregnancies were collected with informed written consent from non-smoking mothers who delivered after 37 weeks’ gestation by elective Cesarean section at the National University Hospital, Singapore between 2018 and 2020 (Table 1). Women were tested for GDM at mid-gestation as part of universal screening by a three time-point 75 g oral glucose tolerance test (OGTT) using WHO 2013 criteria [27]. Placentas were collected from ten normoglycemic and nine GDM pregnancies and matched for first trimester BMI so that cases were balanced across a range of maternal BMIs in both the GDM and non-GDM populations (Table 1). Hence, neither fasting glycemia nor post-load glycemia (by OGTT) were significantly associated with BMI in our study population. All selected GDM pregnancies fulfilled only the criteria of high post-prandial glycemia (1 h ≥ 10.0 mmol/L and/or 2 h ≥ 8.5 mmol/L) and had normal fasting glycemia (<5.1 mmol/L) in order to limit heterogeneity within our GDM study group and reflect the predominant characteristics of GDM in our local Singapore population [28]. Cases of known pre-existing type 1 and type 2 diabetes mellitus were excluded, as were cases of possible pre-existing diabetes defined by antenatal OGTT results of 2 h glycemia ≥11.1 mmol/L. Only one of the GDM cases was treated with insulin, with the remainder diet-controlled.

### 2.1. Placenta Collection and Placental Explant Culture

Placental explants were cultured as previously described [30]. Briefly, five pieces of villous placental tissue were biopsied from across the placenta then cut into small explants (~3 mm^3^). Explants were then cultured for 48 h in serum-free CMRL media containing 1.5% fatty acid free BSA and 24 µM ^13^C stable isotope labelled arachidonic acid (1,2,3,4,5–^13^C), alongside a range of glucose and myo-inositol treatments described in Table 2.

Three replicate wells each containing five explants and 1.8 mL media were conducted for each treatment. The control condition contained physiological amounts of glucose (5 mM) whilst 10 and 17 mM glucose treatments represent GDM-like conditions and a supraphysiological glucose concentration similar to those used in other placental trophoblast studies. Placental explants were treated with 60 µM myo-inositol to reflect the myo-inositol supplied to the placenta from mother and fetus when the mother is supplemented with myo-inositol. Un-esterified AA has previously been reported to be 12.7 (SD 2.1) µM in the maternal vein and 44.2 (5.2) µM in the placental intervillous space [31]. We therefore added 24 µM of ^13^C-AA in our experiments, which was high enough to enable quantification of a range of ^13^C-AA placental lipids. ^13^C-AA is metabolized identically to naturally occurring ^12^C-AA, but is not naturally present in placenta. Therefore, the abundance of ^13^C-AA-lipids in placental explants in vitro reflects the capacity of the placenta to take up AA and metabolize AA-lipids separately from the influence of maternal diet, and maternal and fetal lipid metabolism. Further experimental and material details can be found in Appendix A. Lipids were extracted as described in Appendix A.

### 2.2. LC-MS/MS Methodology

An LCMS method was developed based on the lipidomics methods of the Baker Institute [32]. A dMRM transition list was developed containing transitions for unlabeled lipids containing ^12^C_20_-AA (^12^C-AA-lipids) and the predicted transitions of stable isotope labeled lipids containing ^13^C_5_-^12^C_15_-AA (^13^C-AA-lipids). Only ^13^C-AA-lipids’ transitions which gave peaks that co-eluted with matching ^12^C-AA-lipids’ transitions, and which were only present in explants treated with ^13^C-AA were incorporated into the final method. Lipid extracts were injected into an Agilent 6490 triple quadrupole (QQQ) liquid chromatography mass spectrometry (LC-MS/MS) instrument and analyzed as described in Appendix A.

### 2.3. Data and Statistical Analysis

LC-MS/MS data was analyzed as described in Appendix A and the amount of each lipid expressed as pmol/mg dry placenta. We also calculated the relative ^13^C-AA lipid amount in each lipid class for each placenta (i.e., **∑** amount ^13^C-AA lipid in lipid class/Σ amount ^13^C-AA lipid in total) to explore the compartmentalization of newly introduced ^13^C-AA into different lipid pools.

Linear regression was then run for each lipid under control conditions (outcome: log_2_ amount or log_2_ relative amount of lipid) with each variable of interest (predictor: maternal BMI, fasting glycemia or post-load 2 h glycemia). Where indicated, multiple linear regression was then performed with mutual adjustments for these variables. The Benjamini-Hochberg method was used to correct for multiple testing to minimize false discovery and statistical significance was set at a two-sided alpha level of *p* < 0.05.

We then considered the effects of glucose and myo-inositol treatment, calculating the log_2_ fold-change in lipid amount for each placenta between each treatment condition (Table 2) and control (5 mM glucose, 0.3 µM myoinositol). A log_2_ fold-change in lipid amount > 0 indicates an increase in ^13^C-AA lipids compared to the control, whilst values < 0 indicate a decrease. Statistical difference between treatment and control for each lipid was tested using a one sample t-test comparing the treatment log_2_ fold-change to the reference control (i.e., 0). The Benjamini-Hochberg method was used to correct for multiple testing (for multiple lipids and multiple treatments) and statistical significance was set at a two-sided alpha level of *p* < 0.05. Alterations in relative ^13^C-AA lipid amount in each lipid class were also calculated using the same methodology.

### 2.4. Analysis of Data from Snap Frozen Placenta

We also analyzed an existing dataset from a separate cohort of 50 snap frozen placental biopsies, where endogenous placental inositol content was quantified by an enzymatic assay [33,34], and lipidomics data were acquired using a triple quadrupole LC-MS/MS methodology (Appendix A). We were specifically interested in comparing endogenous placental AA lipid data to our in vitro work. Therefore, data was only extracted for lipids that were also analyzed as part of our placental explant work.

Linear regression was then run for each lipid with each variable of interest (predictor: maternal BMI, fasting glycemia or post-load 2 h glycemia). In order to determine to what extent variations in endogenous ^12^C-AA-lipids may depend on variations in placental inositol content, these models were then re-rerun with the inclusion of inositol content in the model. The Benjamini-Hochberg method was used to correct for multiple testing and statistical significance was set at a two-sided alpha level of *p* < 0.05. Further information on this data set and its analysis can be found in Appendix A.

## 3. Results

### 3.1. Placental Incorporation of Exogenous ^13^C-AA into ^13^C-AA Labeled Lipids

Placental explants incubated with stable-isotope ^13^C-AA for 48 h produced stable-isotope labeled ^13^C-AA-lipids of which seventeen could be reliably quantified using our LCMS method. Quantifiable ^13^C-AA lipids included five phospholipids [phosphatidyl-ethanolamines (PE-38:4 [18:0_20:4]) phosphatidylethanolamine plasmalogen (PE-P 36:4 [16:0_20:4], PE-P 38:4 [18:0_20:4]), phosphatidyl-choline (PC 40:8 [20:4_20:4]; phosphatidyl inositol PI 38:4 [18:0_20:4])], one lyso-phospholipid [lyso-phosphatidyl-choline (LPC 20:4)] and eleven triacylglycerides (TGs) (Figure 1). While the bulk of endogenous ^12^C-AA lipids were PEs (32.8%) and PE-Ps (35.4%), for freshly produced ^13^C-AA lipids most were TGs (75.1%), with less found in PEs (2.8%), PE-Ps (6.3%) and PIs (5.9%) (Figure 1). This indicates that TGs made up a much greater proportion of newly synthesized AA lipids, while PE, PE-P and PI phospholipids were produced more slowly consistent with previous findings for other fatty acids (Palmitic acid: PA, Oleic acid: OA and Docosahexaenoic acid: DHA) [30,35,36].

### 3.2. Association of Maternal Glycemia and BMI with Placental ^13^C-AA Incorporation under Culture with Physiological Glucose (5 mM)

The absolute amount of newly synthesized ^13^C-AA lipids in placental explants was not associated with prior in vivo exposure to the maternal metabolic factors of BMI or glycemia or GDM status for any ^13^C-AA lipid (Figure 2A). However, the proportions of ^13^C-AA lipids in different lipid classes relative to total ^13^C-AA lipids demonstrated distinct patterns of associations with maternal metabolic factors. Increasing maternal fasting glycemia across a normal gestational range (4.1–4.7 mM) was associated with a lower proportion of ^13^C-AA phosphatidyl-ethanolamines (both PE and PE-P) with a similar trend for PI, but positively associated with the proportion of ^13^C-AA TGs (Figure 2B). These findings indicate a shift in AA lipid metabolism away from PE, PE-P and PI and towards TG with increasing fasting glycemia. Adjustment for maternal BMI did not materially change the result (Figure 2). No associations were observed between AA-lipid class proportions with either maternal BMI or 2 h glycemia.

Within the same placental explants, the absolute amounts of unlabeled endogenous ^12^C-AA-lipids (matching our previously discussed ^13^C-AA-lipids), demonstrated associations with maternal glycemia and BMI. Amounts of unlabeled ^12^C-AA PE 38:4 and PI 38:4 were negatively associated with fasting glycemia, with similar trends for ^12^C-AA PE-P. Meanwhile, amounts of ^12^C-AA TG 54:7 was negatively associated with 2 h glycemia, and ^12^C-AA TG 58:9, TG 58:8, TG 56:8 and PE-P 38:4 were negatively associated with maternal BMI (Figure 2C). Although amounts of newly synthesized ^13^C-AA lipids showed hints of corresponding directional trends, significant associations were only observed for ^12^C-AA lipids. Quantified ^12^C-AA lipids were synthesized both during pregnancy and under explant culture, whilst quantified ^13^C-AA lipids were only synthesized during culture. Differences could thus be due to the relatively short period of culture (48 h) and hence a limited ability to generate large amounts of slowly synthesized ^13^C-AA phospholipids. Alternatively, differences could be due to other in vivo factors such as maternal lipid supply that would affect ^12^C-AA-lipids but not in vitro ^13^C-AA-lipid metabolism.

When the proportion of ^12^C-AA lipids (relative to total ^12^C-AA lipids matching quantifiable ^13^C-AA-lipids) was analyzed (Figure 2D), as for ^13^C-AA-lipids, increasing maternal fasting glycemia was positively associated with the proportion of ^12^C-AA TGs. However, maternal fasting glycemia was only negatively associated with the proportion of PE, after adjusting for maternal BMI. BMI itself was positively associated with the proportion of ^12^C-AA PE even after adjusting for fasting glycemia.

### 3.3. Effect of In Vitro Glucose Treatment on Placental ^13^C-AA Lipid Metabolism

Next, we investigated if glucose could directly impact placental ^13^C-AA-lipid metabolism during in vitro explant culture, by measuring the amounts of each AA-lipid after 48 h, which reflects the net effects of lipid synthesis and catabolism. Compared to controls (5 mM glucose), placental explants treated with 10 mM glucose (representing GDM-like conditions, Figure 3A), showed decreases in the amount of all eleven TGs and all PE-P and PI, while ^13^C-AA-PC and ^13^C-AA-LPC were unchanged. The mean difference compared with control ± SD, averaged for each lipid class, was as follows ^13^C-AA-PE (12.3% ± 5), ^13^C-AA-PE-P (25% ± 11), ^13^C-AA-PI (29% ± 13) and ^13^C-AA-TG (28% ± 13). Meanwhile, placental explants treated with supraphysiological 17 mM glucose, showed increases in amounts of only ^13^C-AA-LPC (by 84% ± 62) and ^13^C-AA-PC (33% ± 15) compared with controls. In vitro glucose treatment thus directly alters placental AA metabolism, but the direction of effect and the lipids influenced depend on the concentration of glucose. In vitro glucose treatment had generally similar effects on the amount of unlabeled ^12^C-AA lipids to those seen with labeled ^13^C-AA lipids (Appendix A). These results suggest that increases in glucose concentration can impact both newly synthesized ^13^C-AA lipids and existing endogenous placental ^12^C-AA lipids over the relatively short time span of 48 h.

Among the ^13^C-AA-lipids, the proportion of ^13^C-AA PC was increased by in vitro 10 mM glucose treatment (Figure 3B) compared with controls. In vitro 17 mM glucose treatment increased the proportion of both ^13^C-AA PC and LPC while decreasing the proportion of PE-P and PI. Thus, in vitro glucose treatment alone could not explain the association between maternal glycemia and altered AA-lipid compartmentalization, and chronic exposure to other in vivo factors associated with glycemia are likely involved in programming this response.

### 3.4. Effect of In Vitro Myo-Inositol or Combined Glucose and Myo-Inositol Treatment on Placental ^13^C-AA Lipids

Myo-inositol has been suggested as an intervention to lower gestational glycemia, to prevent GDM and lower the risk of adverse effects in pregnancy and the offspring [37]. We therefore investigated the effects of in vitro myo-inositol treatment on placental ^13^C-AA metabolism and whether myo-inositol could attenuate the changes caused by glucose treatment. Treatment with myo-inositol alone (60 µM, at physiological 5 mM glucose) did not change the amount of ^13^C-AA-lipids after 48 h of culture compared with controls (no additional myo-inositol, 5 mM glucose) (Figure 3A). Furthermore, unlike treatment with glucose, treatment with combined glucose-myo-inositol (10 mM–60 µM or 17 mM–60 µM) also did not change the amount of most ^13^C-AA-lipids compared with controls (Figure 3A). Myo-inositol thus suppressed glucose-induced increases and glucose-induced decreases in placental ^13^C-AA-lipids. The exception was ^13^C-AA PE-P-38:4 which was decreased relative to controls with both glucose alone (10 mM) and glucose-myo-inositol (10 mM–60 µM). Overall, myo-inositol appeared better able to attenuate glucose-induced decreases in TG than phospholipids (PE-P, PE and PI). Similar effects were also observed with ^12^C-AA lipids (Appendix A). Thus myo-inositol, by suppressing some glucose-induced alterations in AA lipid metabolism, whilst allowing others, may be an important regulatory factor in how glucose affects placental AA lipid metabolism.

When considering the proportions of ^13^C-AA-lipids, compared with controls, treatment with myo-inositol alone significantly reduced the proportion of ^13^C-AA-PE-P relative to total ^13^C-AA lipids suggesting that myo-inositol itself may decrease compartmentalization into PE-P (Figure 3B). Meanwhile, the glucose-induced increase in proportions of ^13^C-AA PC and LPC were attenuated with combined glucose/myo-inositol treatment, while the proportions of ^13^C-AA PE-P and PI were further reduced. Thus, while myo-inositol attenuated most glucose-induced changes, decreases in the amount and proportion of ^13^C-AA PE-P and the proportion of ^13^C-AA PI remained largely unresolved, suggesting particular challenges in suppressing these specific glucose-induced alterations. Such findings may relate to why these particular endogenous phospholipids appear decreased with increasing fasting glycemia (Figure 2).

### 3.5. Association of Prior Maternal Fasting Glycemia Exposure In Vivo with Alterations in Placental AA Lipids Induced by Glucose and Myoinositol Treatment In Vitro

Our previous work suggested that the maternal metabolic environment in utero programs the response of the placenta to in vitro treatments [30,38]. We therefore tested for associations between maternal BMI, fasting glycemia and 2 h glycemia, with the subsequent alterations in ^13^C-AA lipids in response to glucose or myo-inositol or combined treatments in vitro.

Prior exposure to increasing fasting glycemia influenced how ^13^C-AA phospholipids were altered by in vitro glucose treatment compared with controls, with significant associations between fasting glycemia and response to 10 mM glucose observed for PC 40:8, PE 38:4, PE-P 36:4, PE-P 38:4 and PI 38:4 (Figure 4-column 3 and Appendix A). Even though in vitro glucose (10 mM) generally induced a decline in most AA-phospholipids (dots below y = 0), the higher the fasting glycemia, the more blunted the glucose-induced decrease (reduced downward distance between the dots and y = 0 (Figure 4, column 3). Indeed, placenta previously exposed to higher than average fasting glycemia even showed increases in AA-LPC and PC with 10 mM glucose treatment; echoing the overall increase in AA-LPC and PC with 17 mM glucose shown in Figure 3. Glucose-induced alterations in ^13^C-AA TGs, in contrast, appeared less influenced by fasting glycemia with significant associations only observed for TG 56:6 (Appendix A). Similar trends were observed with 17 mM glucose-induced alterations, but significant associations were only observed for PE 38:4 (Appendix A) suggesting that very high glucose treatment may override prior glycemia-induced programming. No associations were found between glucose-induced alterations in ^13^C-AA lipids and prior exposure to increasing 2 h glycemia. Overall, these findings suggest that increasing levels of fasting glycemia in prior in vivo exposure may program the placenta to blunt its response to glucose.

Meanwhile, associations between myo-inositol-induced alterations in ^13^C-AA lipids and prior exposure to increasing fasting glycemia were significant for most ^13^C-AA phospholipids (LPC 20:4, PE 38:4, PE-P 36:4, PE-P 38:4, PI 38:4), but were only significant for one TG (TG 50:4, Appendix A). With prior exposure to lower maternal fasting glycemia, myo-inositol treatment generally decreased newly synthesized ^13^C-AA phospholipids (dots below y =0) compared with controls (0.3 µM myo-inositol, 5 mM glucose). In contrast with prior exposure to higher maternal fasting glycemia myo-inositol treatment generally increased ^13^C-AA phospholipids (dots above y =0) (Figure 4, column 2). The association crossed the x axis around a fasting glycemia level of 4.4–4.5 mM, which is the median concentration in the general obstetric population [39]. Since endogenous phospholipids were lower with increasing fasting glycemia exposure (Figure 2C), this suggests that myo-inositol could act to bring AA-phospholipids towards physiological means, thus moderating AA placental metabolism.

The influences of prior fasting glycemia on ^13^C-AA phospholipid alterations with combined glucose-myo-inositol treatment (10 mM/ 60 µM, or 17 mM/60 µM; Figure 5B) were similar to those of myo-inositol (60 µM with 5 mM physiological glucose). Prior exposure to increasing 2 h glycemia or increasing BMI did not influence myo-inositol-induced alterations. Findings were also similar for alterations in unlabeled ^12^C-AA lipids (Appendix A).

To explore the moderating effect of myo-inositol, we further assessed whether myo-inositol could indeed moderate the basal placental AA-lipid metabolism represented by the amount of ^13^C-AA-lipid in the control (0.3 µM myo-inositol, 5 mM glucose). We found that in vitro myo-inositol treatment (60 µM) generally increased ^13^C-AA lipids in explants from placenta which basally produced less ^13^C-AA-lipid, but decreased ^13^C-AA lipids in placenta that basally produced more ^13^C-AA-lipid (Figure 5). These findings were most prominent for phospholipids as reflected by the significant negative associations observed between the amount of ^13^C-AA-lipid in the control and the myo-inositol-induced alterations in ^13^C-AA LPC 20:4, PE 38:4, PE-P 36:4, PE-P 38:4, PI 38:4 and TG 50:4, with similar trends for remaining AA-lipids. Findings were similar for alterations in unlabeled ^12^C-AA lipids (Appendix A).

Overall, these findings suggest that myo-inositol may moderate variations in basal placental AA metabolism, shifting them towards a physiological mean. All of these may constitute adaptive responses in placental AA metabolism to cope with prevailing levels of maternal glycemia or other factors influencing placental AA metabolism.

### 3.6. Supporting Evidence from a Separate Cohort: Endogenous ^12^C-AA Lipids in Snap-Frozen Placental Biopsies

We sought to replicate some of our findings using an existing ex vivo dataset from a separate cohort of 50 placental biopsies (snap frozen within 30 min of cesarean delivery) with measurements of endogenous placental inositol content (quantified by an enzymatic assay) and lipidomic data (Appendix A). Data on ^12^C-AA-lipids matching those in our current study was extracted to determine whether endogenous inositol may influence the relationship between maternal glycemia, and the absolute and proportional (relative to total ^12^C-AA lipids of interest) amounts of endogenous ^12^C-AA-lipids.

Maternal glycemia was positively associated with the amount of several placental ^12^C-AA TGs (with fasting glycemia: TG 54:6, TG 54:7, TG 58:8; with 2 h glycemia: TG 54:5, TG 54:7, TG 56:6, TG 56:7 and TG 58:8) (Figure 6A). Both fasting and 2 h glycemia also positively associated with the proportion of endogenous ^12^C-AA-TGs, but negatively associated with the proportion of ^12^C-AA-PE-P and PI (Figure 6B), similar to patterns for ^13^C-AA lipids observed in placental explants following 48 h culture under control conditions (Figure 2). Since ^13^C-AA lipids represent only the lipids that were synthesized during the 48 h culture in the absence of on-going maternal-fetal influences, these similar variations in ^12^C-AA in ex vivo biopsies are likely also due to variations specifically in placental processing and metabolism rather than in the supply of AA from maternal sources or utilization by the fetus.

As increasing glycemia is known to associate with reduced endogenous inositol in placenta [33], we explored the consequences of neutralizing such an influence by statistically adjusting for placental inositol. Adjusting for endogenous placental inositol content in the snap frozen placenta dataset, attenuated associations between maternal glycemia (fasting and 2 h) and the amount and proportion of AA lipids (Figure 6B). Consistent with our in vitro data, this suggests that variations in in vivo endogenous placental myo-inositol may regulate the relationship between maternal glycemia and endogenous placental AA-lipids. No associations were seen between placental AA-lipids and maternal BMI.

## 4. Discussion

Our findings suggest increasing maternal fasting glycemia programs the placenta to decrease the compartmentalization of AA into placental phospholipid pools, but increase compartmentalization into TGs. Meanwhile in vitro glucose treatment directly decreases the abundance of most freshly synthesized placental AA lipids, with most of these effects attenuated by myo-inositol treatment. Myo-inositol treatment alone also moderated basal placental AA-lipid processing towards physiological means. Overall, these observations suggest that myo-inositol promotes adaptive responses to overcome the effects of glucose on placental AA lipid metabolism. This notion is consistent with analysis of endogenous AA lipid data from a separate cohort, where endogenous myo-inositol also appears to modulate glycemia-induced placental AA-lipid alterations in vivo.

### 4.1. Maternal Glycemia and Placental AA-Lipid Compartmentalization and Bioavailability

The compartmentalization of AA into different lipid pools (Figure 7) is key to its bioavailability, its utilization and to placental function and signaling [40,41,42]. Placental AA-phospholipids are thought to be the source of fetal AA supply, and the source of AA for eicosanoid synthesis for paracrine and endocrine signaling [43,44,45]. Meanwhile, placental AA-TGs act as AA storage pools [44,46]. Thus, our finding of a glycemia associated shift in placental compartmentalization away from PEs, PE-Ps and PI and thus away from AA pools utilized for fetal supply and eicosanoid synthesis, could potentially impact placental function and fetal development.

Our findings are consistent with a study [47] of perfused placentas from women with pre-existing diabetes showing increased radiolabeled placental ^14^C-AA TG and decreased ^14^C-AA phospholipids alongside decreased export of un-esterified ^14^C-AA to the fetus compared with non-diabetic placentas. Altered compartmentalization could thus potentially explain the reduction in fetal circulating non-esterified AA in maternal diabetes [5,6,7]. We next investigated whether such associations could be partly explained by a direct effect of high glucose on placental AA-metabolism.

### 4.2. Glucose-Induced Changes in Placental AA-Lipids In Vitro

There was a general decrease in all classes of freshly synthesized ^13^C-AA-containing lipids with 10 mM in vitro glucose treatment suggesting impact on an upstream process such as decreased placental AA uptake and/or activation. One possibility is AA-CoA synthesis, which prevents re-export into the maternal circulation and which is an essential first step for all lipid synthesis. This would be consistent with studies in isolated human trophoblasts from GDM pregnancies demonstrating less radiolabeled ^14^C-AA accumulation and decreased *ACSL1* (acyl-CoA synthetase) mRNA expression compared to non-GDM controls [48].

However, with 17 mM glucose, most lipids were left unaltered compared to controls, whilst ^13^C-AA-PC and ^13^C-AA-LPC were actually increased. This lack of change with very high glucose mirrors published findings where 20 mM glucose was found to have little impact on overall ^14^C-AA uptake by primary cytotrophoblasts [48]. One possible explanation for increased PC and LPC may be increases in the activity of phosphocholine cytidylyltransferase or phosphatidylethanolamine N-methyltransferase, which catalyze the rate limiting steps in PC synthesis, and which are known to be upregulated by hyperglycemia or glucose treatment in a range of tissue types [49,50,51]. However, alterations in these genes would not explain why other lipids are decreased by 10 mM but not 17 mM glucose.

Decreases in the proportion of PE and PE-P both in association with increasing fasting glycemia and by in vitro 10 mM glucose treatment could also suggest that glucose may specifically increase catabolism of PE-P and PI, but not LPC and PC. This could be mediated by phospholipase PLA2G2A, which is abundant in the placenta [52,53,54], is known to be increased in GDM [55,56] and which in non-placental tissue has particular specificity for AA-PE over AA-PC [57,58].

### 4.3. Placental Programming by Prior Maternal Glycemia

Our findings suggest that prior exposure to increasing fasting glycemia blunts the effects of in vitro glucose treatment on placental AA-phospholipid metabolism, potentially representing an adaptive response to protect the placenta and fetus from high glucose. For example, decreased placental lipid synthesis or increased PE and PE-P catabolism could make more un-esterified AA available for fetal transfer even in a low AA environment. However, any alterations in placental lipid metabolism could have both positive and negative effects. For example, glucose-induced increases in placental AA-LPC could increase net export of AA to the fetus in the AA-LPC form to compensate for decreased supply of AA in the classical un-esterified form. This would be consistent with reports of increased transfer of an unidentified AA-containing phospholipid species, from perfused diabetic placenta into the fetal effluent compared to non-diabetic controls [47]. However, a possible trade-off of this form of compensation could be an increased risk of macrosomia since many LPCs are known to promote fetal growth [59,60,61], which would further exacerbate the direct effects of increased transplacental glucose and fetal hyperinsulinemia.

### 4.4. Myo-Inositol as a Moderator of Placental AA Metabolism

Prior exposure to increasing maternal fasting glycemia across the normal range also influenced the effects of in vitro myo-inositol on AA-lipid metabolism, with myo-inositol bringing basal AA-lipid amounts towards a physiological mean. Thus, myo-inositol likely plays a regulatory rather than direct role; influencing metabolic pathways that are programmed by glucose and other factors in order to moderate placental lipids. Prior exposure to higher maternal glycemia may also alter the effects of myo-inositol treatment by reducing inositol uptake and synthesis [33].

Lipid metabolism involves many pathways that use inositol-containing signaling molecules. For example, the phosphorylation of phosphatidyl inositol diphosphate (PIP2) into phosphatidyl inositol triphosphate (PIP3) activates protein kinase B, an enzyme which upregulates proteins important for fatty acid uptake [62]. Meanwhile the cleavage of PIP2 activates protein kinase C [63,64] which inhibits ACSL4 (an enzyme important for AA uptake and lipid synthesis [65,66]) and stimulates PLA2 (which is important for AA-lipid catabolism [63,67,68]). Both PIP2-related regulatory pathways are impaired in the placenta and in other tissues with hyperglycemia [64,69,70,71]. If such impairment is due to a deficiency in inositol-containing signaling molecules, increasing myo-inositol may attenuate many of these changes.

Myo-inositol appeared more able to attenuate glucose-induced decreases in AA-TG than AA-phospholipids. Differential regulation of lipids by myo-inositol has also previously been reported in the livers of myo-inositol deficient rats where AA-TGs were decreased but phospholipids remained unchanged [72]. However, myo-inositol in our study also appeared less able to attenuate glucose-induced decreases in PE-P 36:4 and PI 38:4 than other phospholipids and did not attenuate PE-P 38:4. These findings suggest the existence of glucose-induced processes targeting PE-P and PI metabolism which either do not involve inositol (nor amenable to regulation by inositol), or where inositol might have a similar effect to glucose. For example, inositol may also increase AA-phospholipid catabolism since lecithin:cholesterol acyltransferase (LCAT), a phospholipase with a preference for AA-phospholipids [73,74], is known to be decreased in the plasma of myo-inositol deficient rats [75,76]. The resistance of PE-P and PI to modulation could also contribute to explaining why maternal fasting glycemia is associated with decreased compartmentalization of AA into these phospholipids since these may be decreased by glucose even when myo-inositol supply is adequate.

There is also the possibility of bidirectional interplay between inositol signaling and AA bioavailability. Previous in vitro experiments reported that AA can stimulate the hydrolysis of placental phosphatidyl-inositol-phosphates to release inositol phosphates and human placental lactogen (hPL), a hormone important for maternal adaption and fetal development [25,77]. Many inositol containing signaling molecules also contain AA, so their abundance will be influenced by AA availability. Indeed, our experiments show that glucose-induced decreases in ^13^C-AA lipids include decreased ^13^C-AA phosphatidyl inositol (PI). Thus, the dysregulation of inositol metabolites and inositol signaling in hyperglycemia should be seen as a consequence not only of inadequate inositol but also dysregulated AA metabolism [17,33].

### 4.5. Clinical Implications

AA released from different phospholipids are directed towards synthesis of particular eicosanoids, with prostaglandins more likely to be synthesized from AA released from PEs than other phospholipids [40,78,79,80]. It is thus plausible that glycemia-induced alterations in AA metabolism underlie pathological changes in eicosanoid production in maternal diabetes and explain why glucose decreases some eicosanoids while increasing the synthesis of others [8,9,10], which may have implications for fetal growth disorders and preterm labor.

It is not clear yet which glucose-induced alterations in placental AA metabolism are involved in hyperglycemia-related pathophysiology. If myo-inositol can suppress pathological glucose-induced changes in AA metabolism during pregnancy, while sparing or enabling desired/beneficial adaptive changes, myo-inositol supplementation could be used to minimize glucose-induced disruptions in placental AA metabolism and potentially reduce hyperglycemia-related pregnancy complications.

## 5. Strengths and Limitations

This work isolated placental AA metabolism itself for study, minimizing the influence of maternal supply (reflective of diet, maternal tissue metabolism and placental blood flow) and fetal utilization. Our work is thus able to demonstrate the impact of maternal glycemia associated programming specifically within placental tissue and the direct effects of in vitro glucose and myo-inositol treatment. However, this work can only be done on cultured term placenta following delivery and therefore does not necessarily represent the in vivo situation earlier in pregnancy. Explant based stable isotope experiments cannot be run on large numbers of placenta, limiting the statistical power to explore concurrently the effects of multiple other factors such as parity, ethnicity and socio-demographics. Our experiments were also limited to Indian and Chinese women in Singapore and results may differ in other populations. GDM in Singapore is generally characterized by an elevated post-prandial glycemia in the OGTT, rather than by high fasting glycemia, with limited numbers of the latter group suitable for recruitment. Thus, our sample only included women with fasting glycemia within clinically-defined normal ranges yet we were able to elicit fasting glycemia-induced variations in placental AA-metabolism consistent with notions that maternal glycemia demonstrates a continuum of effect across the glycemic range [26]. However, whether our AA results continue to follow a similar linear trajectory with fasting glycemia above the levels studied is unclear. Most GDM cases (8 out of 9) were diet controlled and it is unknown whether results would be altered by insulin or metformin treatment in pregnancy.

Larger cohort studies will be needed to determine if inositol supplementation pre-pregnancy and during gestation could change placental AA metabolism to impact maternal and fetal outcomes such as preterm birth. Placental perfusion studies may be useful in determining specifically how inositol may influence glucose-induced changes in transplacental nutrient transport and placental signaling.

## 6. Conclusions

Prior exposure to different levels of fasting glycemia was associated with variations in how the placenta metabolizes AA, while in vitro evidence suggested that glucose directly alters some placental AA metabolic pathways. Since well-regulated placental AA-lipid metabolism is critical for maternal and fetal health, glucose-associated alterations in placental AA metabolism may negatively impact maternal-placental-fetal signaling, fetal growth and development, and parturition.

Myo-inositol moderated AA metabolism towards the mean and attenuated most glucose-induced alterations in AA metabolism. If myo-inositol is also able to moderate placental AA-lipid metabolism during pregnancy, including the effects induced by maternal hyperglycemia, then optimizing placental inositol content may physiologically play an important regulatory role for healthy fetal development and parturition. Thus, in conditions where placental inositol is decreased such as gestational diabetes, maternal inositol supplementation may be useful. Further studies are required to understand the role of inositol in maternal-placental-fetal physiology and whether maternal inositol supplementation is beneficial in pregnancy.

## Figures and Tables

**Figure 1 nutrients-14-03988-f001:**
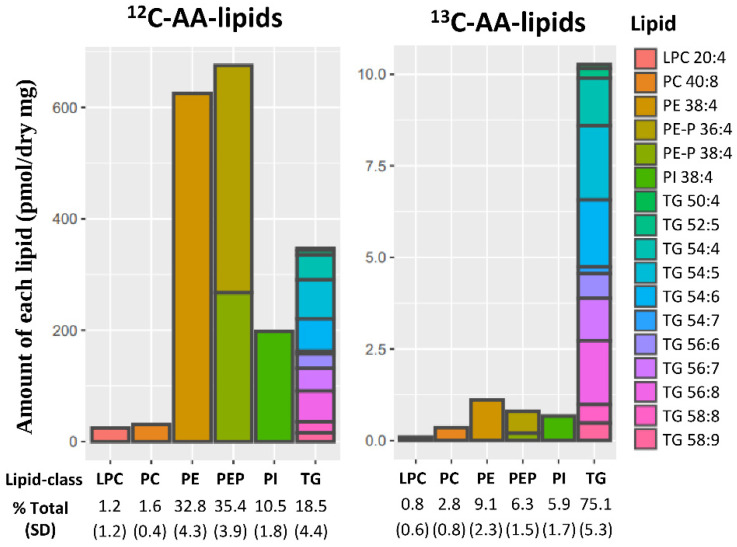
Placental lipids containing endogenous ^12^C-AA or newly synthesized stable isotope labeled ^13^C-AA in placental explants from 19 placenta incubated for 48 h with 24 µM ^13^C-AA under physiological glucose conditions (5 mM). Amount of each lipid (pmol/mg dry placenta) with percentages showing relative amount of each lipid class compared to total ^12^C-AA or ^13^C-AA lipids respectively. Brackets show standard deviation (SD). Colors represent individual lipids. AA: arachidonic acid, LPC: lyso-phosphatidylcholine, PC: phosphatidyl-choline, PE: phosphatidyl-ethanolamine, PE-P: PE-plasmalogen, PI: Phosphatidylinositol, TG: triacylglycerols.

**Figure 2 nutrients-14-03988-f002:**
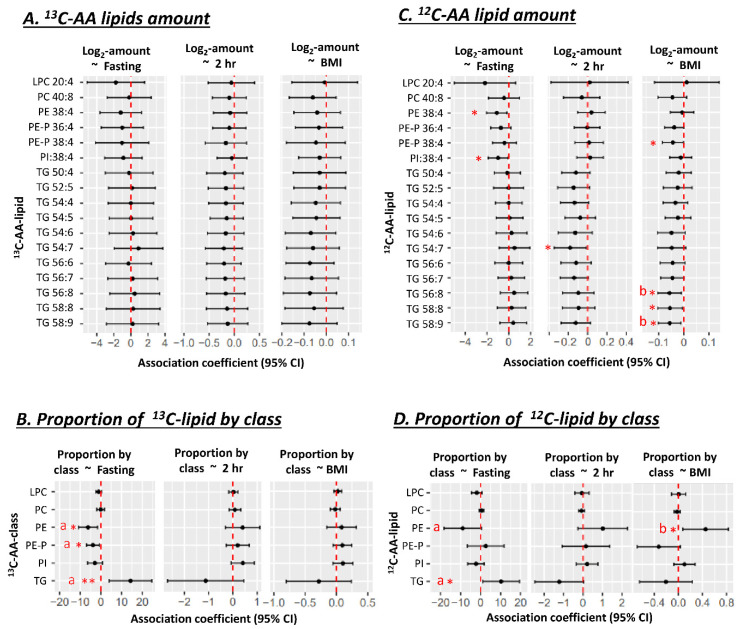
Associations of maternal glycemia and BMI with ^13^C or ^12^C AA-lipids in placental explants (*n* = 19) after 48 h of culture in a physiological glucose concentration (5 mM). (**A**,**B**) Associations with lipid amount (pmol/dry mg placenta). (**C**,**D**) Associations with the proportions of ^13^C or ^12^C AA-lipids relative to total quantified ^13^C or ^12^C AA-lipids. Proportion of ^13^C-lipid in each class = 100 × Sum amount of ^13^C-lipid in lipid class/total quantified ^13^C-lipid in all classes. Results for ^13^C-AA-lipids and corresponding ^12^C-AA lipids are from the same placental explants. Linear regression was run with lipid amount or proportion as the outcome and glycemia (in mmol/L) or BMI (in kg/m^2^) as the predictor. Forest plots show association coefficients and 95% confidence intervals. * shows significant *p* < 0.05. ** shows significance at *p* < 0.01. “a” shows significant *p* < 0.05 for glycemia after adjusting for maternal BMI. “b” shows significance with BMI *p* < 0.05 after adjusting for maternal glycemia. The Benjamini-Hochberg (BH) method was used to correct for multiple testing. Abbreviations—AA: Arachidonic acid, LPC: lyso-phosphatidylcholine, LPE: lyso-phosphatidylethanolamine, PC: phosphatidylcholine, PE: phosphatidylethanolamine, PE-P: phosphatidylethanolamine-plasmalogen, TG: triacylglycerol.

**Figure 3 nutrients-14-03988-f003:**
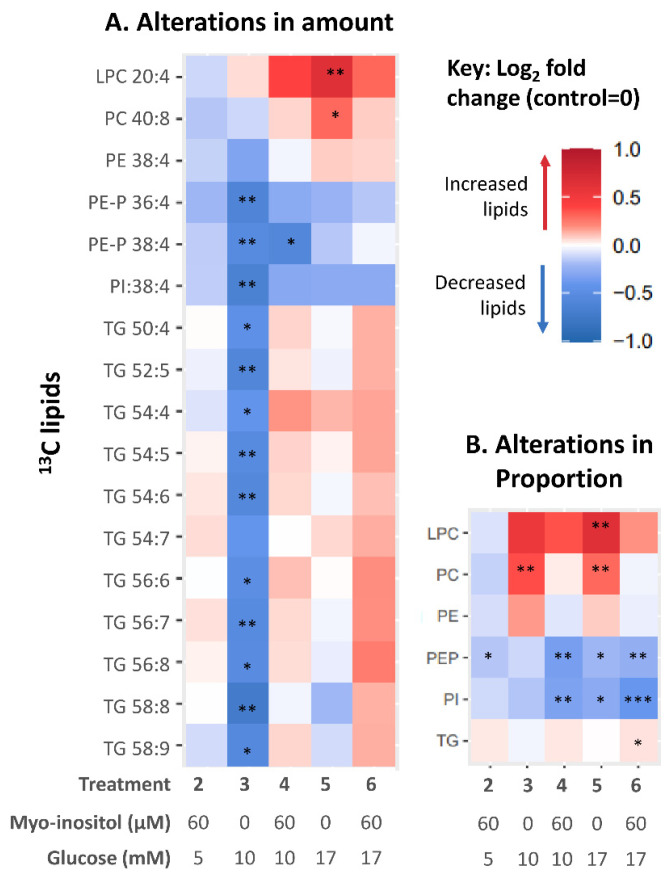
Heat map illustrating alterations in ^13^C-AA lipids in placental explants in response to glucose and myo-inositol treatment. Color indicates the relative change (Log_2_-fold change) in AA lipid in placental explants treated with myoinositol (60 µM) and/or glucose (10 or 17 mM) compared with controls from the same placenta treated with no additional glucose (5 mM) or myo-inositol (0.3 µM). (**A**) Alterations in amount compared to control; (**B**) Alterations in the proportion of each AA lipid class compared to control. Asterisks indicate significant differences by one sample *t* test compared to control (test value = 0) after Benjamini-Hochberg correction for multiple comparisons * *p* < 0.05, ** *p* < 0.01, *** *p* < 0.001.

**Figure 4 nutrients-14-03988-f004:**
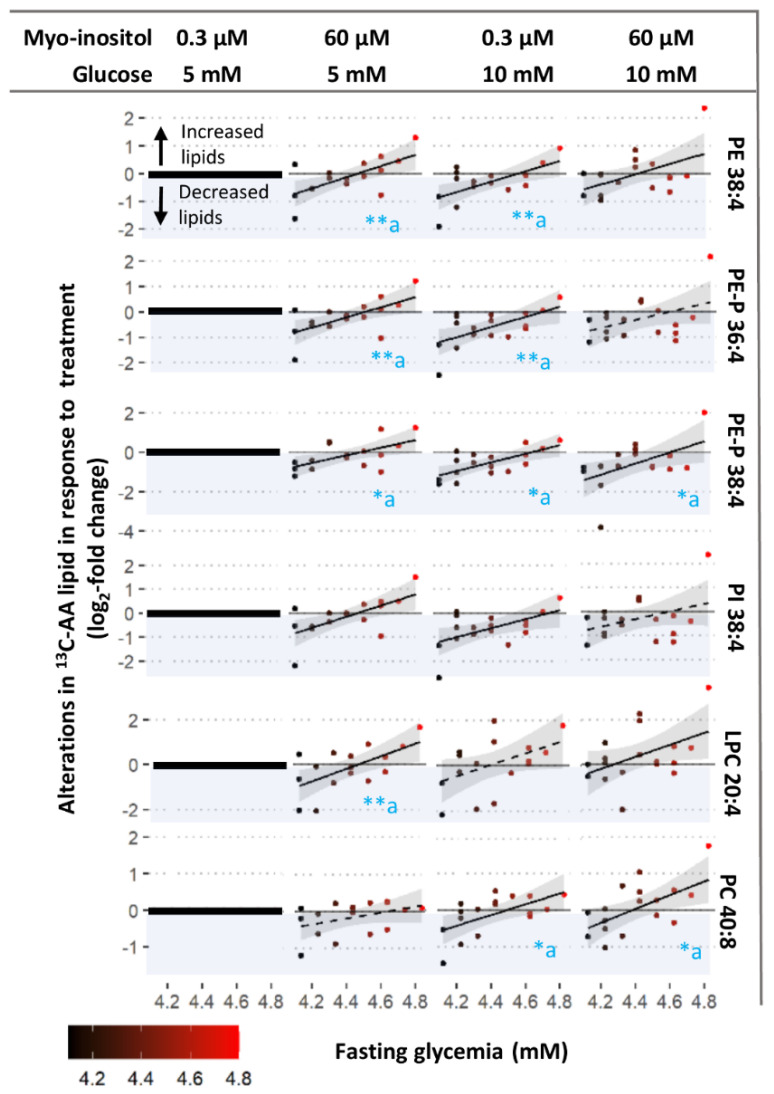
Associations between maternal fasting glycemia and degree of alterations in ^13^C-AA lipid in response to treatment with glucose (10 mM), myo-inositol (60 µM) and combined glucose and myo-inositol. Alterations in ^13^C-AA-lipids expressed in log_2_-fold change relative to amount of ^13^C-AA lipid in control explants (0.3 µM myo-inositol and 5 mM glucose) from the same placenta (zero on y-axis). Positive log_2_-fold change values indicate an increase in ^13^C-AA lipids compared to the control, whilst negative values (blue shading) indicate a decrease. Linear regression was run with log_2_-fold change in lipids as the outcome and fasting glycemia as the exposure variable. Asterisks indicate significant associations * *p* < 0.05, ** *p* < 0.01, “a” indicates that the association remains after adjusting for maternal BMI. The Benjamini-Hochberg (BH) method was used to correct for multiple testing. Solid lines show significant associations while dashed lines show non-significant trends, shaded areas show 95% confidence intervals.

**Figure 5 nutrients-14-03988-f005:**
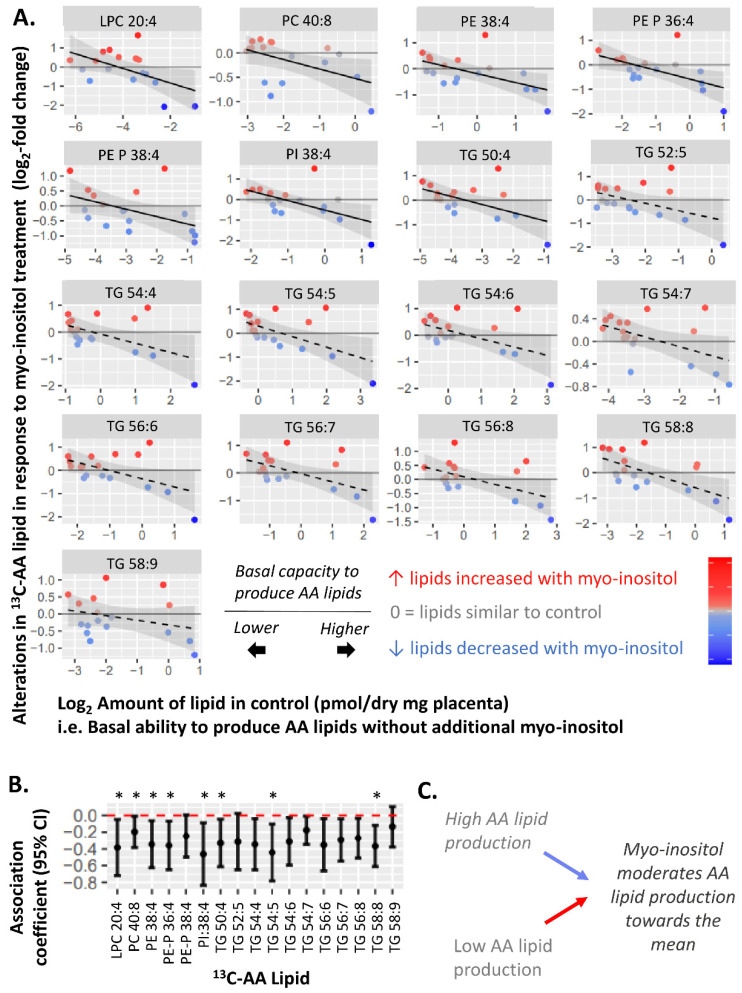
Negative associations between amount of each ^13^C-AA lipid in the control and alterations in ^13^C-AA lipid (log_2_-fold change) in response to myo-inositol treatment. Myo-inositol-induced alterations represent the relative amount of ^13^C-AA lipid in placental explants treated with myo-inositol (60 µM myo-inositol, 5 mM glucose) compared to control explants (0.3 µM myo-inositol, 5 mM glucose) from the same placenta. Positive log_2_-fold-change values indicate an increase in ^13^C-AA lipids compared to the control, whilst negative values indicate a decrease. Linear regression was run with myo-inositol response as the outcome and fasting glycemia as the exposure variable. The Benjamini-Hochberg (BH) method was used to correct for multiple testing. (**A**) Solid lines show significant associations while dashed lines show non-significant trends, shaded areas show 95% confidence intervals. (**B**) Forest plot showing association coefficients and 95% confidence intervals. Asterisks indicate significant associations * *p* < 0.05. (**C**) Overall, these findings suggest that myo-inositol may moderate variations in basal placental AA metabolism, shifting them towards a physiological mean.

**Figure 6 nutrients-14-03988-f006:**
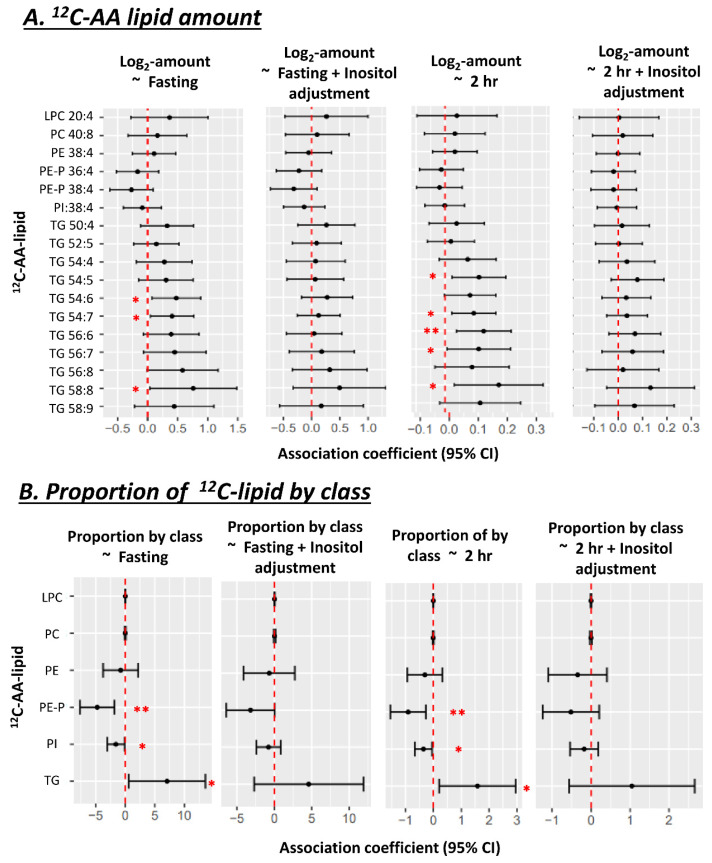
Associations between amount (**A**) and proportion (**B**) of ^12^C-AA lipids with maternal glycemia, before and after adjusting for total placental inositol content, in uncultured snap-frozen placental biopsies (*n* = 50). ^12^C-AA-lipids were extracted from this existing dataset to match ^13^C-AA lipids measured in our placental explant study. Proportion of ^12^C-AA-lipids in each class = 100 × Sum amount of ^12^C-lipid in lipid class/total quantified ^12^C-lipid in all classes. Linear regression was run with lipid proportion as the outcome and fasting glycemia as the predictor. Forest plots show association coefficient and 95% confidence intervals for each lipid class. Asterisks indicate significant associations * *p* < 0.05, ** *p* < 0.01. The Benjamini-Hochberg (BH) method was used to correct for multiple testing.

**Figure 7 nutrients-14-03988-f007:**
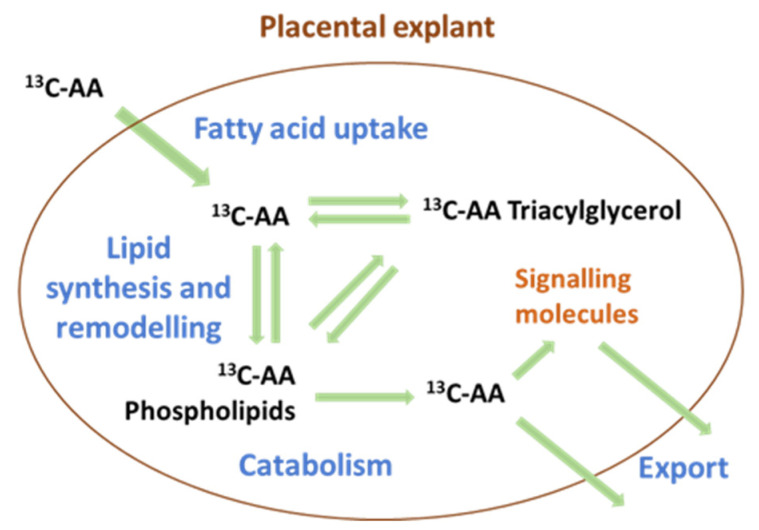
The compartmentalization of arachidonic acid (AA) into different lipid pools regulates both the uptake and export of AA. AA-triacylglycerols often act as storage, while AA-phospholipids are the primary source of bioactive signaling molecules such as eicosanoids and of free AA for export to the fetus.

**Table 1 nutrients-14-03988-t001:** Clinical characteristics of study population.

Clinical Characteristics	Total (*n* = 19)	Non-GDM (*n* = 10)	GDM (*n* = 9)
Maternal Age (years)	32.7 (2.4)	32.4 (2.3)	33 (2.6)
Chinese:Indian ethnicity	12 (63%), 7 (37%)	6 (60%), 4 (40%)	6 (67%), 3 (33%)
Maternal BMI in first trimester (kg/m^2^)	25.2 (5.0)	25.4 (5.5)	25 (4.8)
Normal weight: Overweight: Obese ^†^	6 (32%), 8 (42%), 5 (26%)	4 (40%), 3 (30%), 3 (30%)	4 (44%), 3 (33%), 2 (22%)
Fasting glycemia (mmol/L) ^#^	4.4 (0.2)	4.4 (0.2)	4.3 (0.2)
1-h glycemia (mmol/L) ^#^	9.0 (1.6)	7.9 (1.3)	10.3 (0.6)
2-h glycemia (mmol/L) ^#^	7.3 (1.6)	6.3 (0.9)	8.4 (1.6)
Gestational age at delivery (days)	271.0 (5.6)	272.8 (6.6)	268.7 (3.4)
Female neonates	9 (47%)	5 (50%)	4 (44%)
Birthweight (g)	3150 (356)	3258 (331)	3032 (365)
Birthweight Centile (%)	52.4 (33)	57.3 (33)	47 (34.8)

Data presented as Mean (SD) or n (%). ^†^ WHO recommended Asian-specific BMI cut-off that is associated with increased metabolic risk [29]: Normal 18.5–22.9 kg/m^2^, Overweight 23–27.4 kg/m^2^, Obese ≥ 27.5 kg/m^2^. ^#^ In mid-gestation 75 g three time-point oral glucose tolerance test. BMI kg/m^2^, Body Mass Index; GDM, Gestational diabetes mellitus.

**Table 2 nutrients-14-03988-t002:** Placental explant culture conditions.

Treatment	1 (Control)	2	3	4	5	6
Glucose (mM)	5	5	10	10	17	17
Myo-inositol (µM)	0.3	60	0.3	60	0.3	60
Number of samples	19	17 ^	17 ^	18 ^	18 ^	18 ^

^ Due to technical issues in processing, a few samples were unable to yield data, hence, n < 19.

## Data Availability

Supporting data can be found in the Appendix A. Further information can be supplied upon request.

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
