# Peer review of "Myo-Inositol Moderates Glucose-Induced Effects on Human Placental 13C-Arachidonic Acid Metabolism"

_nutrients, 2022, doi:10.3390/nu14193988_

Round 1

Reviewer 1 Report

This study describes differences in arachidonic acid metabolites in placentas from control and GDM pregnancies, and compares their AA metabolism in vitro, under physiological, elevated and supraphysiological glucose conditions. It also examined how a proposed treatment for GDM metabolic disturbances, myo-inositol, influences AA metabolism under these conditions. The effect of myo-inositol is also estimated by examining myo- inositol concentrations along side AA metabolite concentrations in previously frozen placental samples.

The paper is very well written, and does a good job summarizing quite complicated results, both in text and clarity of figures.  The work is comprehensive and elegant, made possible by the use of stable isotope tracing, which not many labs in this field can do. The detailed supplementary methods are really helpful for readers.

The interpretation of Line 425/figure 6B doesn’t make sense to me.  Adjusting for inositol in the regression model attenuated the relationship between glycemia and AA lipids. The paper suggests this is consistent with the result from in vitro data that treating with myo-inositol attenuated AA differences. But I would think the opposite is true: if myo-inositol attenuates AA differences, then shouldn’t (mathematical) adjustment for myo-inositol concentrations make the differences appear larger? I.e this lipid concentration would be even more different if it weren’t for myo-inositol? If adjusting for myo-inositol attenuates differences, then it was contributing to the difference.

Otherwise, just a few minor suggestions/observations:

    1.       Though they were not significantly different, the clinical characteristics in Table 1 would be better presented in separate columns for the GDM and control groups (or separate columns plus combined), with results of statistical comparisons.  In other words, formatted more like supplementary 5.2.

      2.       It would be nice to have an overview diagram of the arachidonic acid metabolic pathways being studied

3.       Paragraph starting 188:  PE-P isn’t defined, though all the other lipid species discussed are spelled out the first time.    PA, OA and DHA also are mentioned but not spelled out.

     4.       Figure 4 legend:  I’m pretty sure that “Asterisks indicate significant associations” refers to the association between fasting glycemia and lipid concentrations, but it isn’t completely clear.

Reviewer 2 Report

This is a very well written manuscript with clearly presented data and conclusions. The work is also of significant merit to the field. Please address the following minor comments:

1. Did the authors compare the how C13-AA was metabolized between their normoglycemic and GDM cohorts? It is understood that for all the analysis in the manuscript the data was normalized to the original explant, but a basal understanding of inherent differences between the normal and GDM cohort would further add to their conclusions.

2. How does the classification of GDM based just on post prandial glucose ft into the widely used 1 step/2 step glucose tolerance test criteria used for GDM diagnosis.

3. Most of the figures are missing the n  values which begs the question if all the 19 samples were plotted and if not, was there a specific criteria used to exclude them.

4. The authors specify that most of cases in GDM cohort (8 out of 9) were diet controlled. This can be highlighted again in the discussion with a comment on how insulin/drug treated GDM might behave differently.
